

# Identifying transgene insertions in *Caenorhabditis elegans* genomes with Oxford Nanopore sequencing

Paula E. Adams[1,2], Jennifer L. Thies[2,3], John M. Sutton[2,4], Joshua D. Millwood[2,5], Guy A. Caldwell[2], Kim A. Caldwell[2] and Janna L. Fierst[6,7]

[1] Department of Biological Sciences, Auburn University, Auburn, AL, United States of America
[2] Department of Biological Sciences, University of Alabama - Tuscaloosa, Tuscaloosa, AL, United States of America
[3] Curriculum in Toxicology and Environmental Medicine, University of North Carolina at Chapel Hill, Chapel Hill, NC, United States of America
[4] Absci, Vancouver, WA, United States of America
[5] Department of Biological and Environmental Sciences, University of West Alabama, Livingston, AL, United States of America
[6] Department of Biological Sciences, Florida International University, Miami, FL, United States of America
[7] Biomolecular Sciences Institute, Florida International University, Miami, FL, United States of America

Corresponding authors
Paula E. Adams,
pea0013@auburn.edu
Janna L. Fierst, jfierst@fiu.edu

## ABSTRACT

Genetically modified organisms are commonly used in disease research and agriculture but the precise genomic alterations underlying transgenic mutations are often unknown. The position and characteristics of transgenes, including the number of independent insertions, influences the expression of both transgenic and wild-type sequences. We used long-read, Oxford Nanopore Technologies (ONT) to sequence and assemble two transgenic strains of *Caenorhabditis elegans* commonly used in the research of neurodegenerative diseases: BY250 (pPdat-1::GFP) and UA44 (GFP and human $\alpha$-synuclein), a model for Parkinson's research. After scaffolding to the reference, the final assembled sequences were ∼102 Mb with N50s of 17.9 Mb and 18.0 Mb, respectively, and L90s of six contiguous sequences, representing chromosome-level assemblies. Each of the assembled sequences contained more than 99.2% of the Nematoda BUSCO genes found in the *C. elegans* reference and 99.5% of the annotated *C. elegans* reference protein-coding genes. We identified the locations of the transgene insertions and confirmed that all transgene sequences were inserted in intergenic regions, leaving the organismal gene content intact. The transgenic *C. elegans* genomes presented here will be a valuable resource for Parkinson's research as well as other neurodegenerative diseases. Our work demonstrates that long-read sequencing is a fast, cost-effective way to assemble genome sequences and characterize mutant lines and strains.

## INTRODUCTION

With the advent of modern long-read DNA sequencing such as Oxford Nanopore Technologies (ONT) and Pacific Biosystems, the creation of high-quality reference genomes

and characterization of the full spectrum of biodiversity is possible with high precision. For model organisms, biodiversity frequently encompasses mutant lines and strains including those created in laboratory settings with transgenic techniques. *Caenorhabditis elegans* is a well-established model organism with robust genomic resources (*Brenner, 1974*; *Nigon & Félix, 2005-2008*; *Brenner, 2009*) including both natural and induced mutant strains. Currently, there are more than 24,000 mutant lines maintained by the *Caenorhabditis* Genetics Center (CGC) alone (*Caenorhabditis Genetics Center , 2024*) with additional strains housed in individual research labs.

Transgenic lines are a valuable research tool (*Jaenisch & Mintz, 1974*), but the location and effects of transgene insertions are often unknown. Many were created with aggressive methods that may induce "insertional mutagenesis" effects, such as changes in expression of genes neighboring the insertion (*Laboulaye et al., 2018*), large-scale insertions and deletions of endogenous DNA (*Goodwin et al., 2019*), or even chromosomal rearrangements (*Maroilley et al., 2023*). Identifying the exact location of transgenes can assist with diagnosing unintended insertional effects that may otherwise complicate experiments using genetically modified organisms.

Identifying the genomic changes underlying transgenic phenotypes has been challenging. A variety of methods have been used to verify insertion sites including Southern blotting (*Southern, 1975*; *Zastrow-Hayes et al., 2015*), polymerase chain reaction (PCR) (*Nain et al., 2005*; *Yang et al., 2005*), targeted DNA microarrays (*Leimanis et al., 2006*), and next-generation sequencing(NGS) (*Liang et al., 2014*; *Wahler et al., 2013*; *Yang et al., 2013*; *Kovalic et al., 2012*; *Guttikonda et al., 2016*; *Park et al., 2017*). However, each of these methods leaves ambiguities regarding the exact location, size and sequence of most transgene insertions (*Pauwels et al., 2015*).

Long read sequencing has been used to successfully sequence transgene insertions in mice (*Suzuki et al., 2020*; *Nicholls et al., 2019*), and genetically modified plant species such as canola, white clover, and perennial ryegrass (*Giraldo et al., 2021*). These studies chose ONT sequencing due to its characteristically low costs and long read lengths (*Suzuki et al., 2020*; *Jain et al., 2016*; *Van Dijk et al., 2018*). They were able to identify insertion, copy number, and other insertional mutagenesis effects including *E. coli* DNA that had contaminated a transgene (*Nicholls et al., 2019*). Each of these studies searched for transgenes in DNA libraries, leaving a potential for ambiguity in genomic characterization.

Here, we assemble the first genome sequences of two transgenic strains of *C. elegans*, BY250 and UA44. *C. elegans* is used as a model for many neurodegenerative diseases such as Alzheimer's, Parkinson's, Amyotrophic lateral sclerosis (ALS), Huntington's, and other movement and dementia related disorders (*Alexander, Marfil & Li, 2014*). BY250 expresses green fluorescent protein (GFP) in its neuronal cells, making it an excellent strain for neurodegeneration work (*Nass et al., 2005*). UA44 expresses human $\alpha$-synuclein along with GFP and is used as an age-related neurodegeneration model (*Cao et al., 2005*; *Hamamichi et al., 2008*).

Chromosome-scale reference genomes exist for some wild-collected strains of *C. elegans* (*Lee et al., 2021*) but virtually none of the mutant lines or transgenic insertions have been characterized, resulting in a potential gap in studies using transgenic lines. For
example, of the 24,000 mutant lines maintained by the CGC, the two strains reported in this study are the first to be sequenced and characterized at a whole genome level, and have their genomes deposited with the National Center for Biotechnology Information (NCBI) for use by the scientific community. BY250 and UA44 are widely used in neurodegeneration research and verification of the insertion sites and copy number along with the new assemble genomes will further work in this field. Additionally, our results illustrate the innovative power of ONT genomic characterization to complement well-established genetic studies in traditional model organisms.

## MATERIALS AND METHODS

Portions of this text were previously published as part of a thesis (*Adams, 2022*; https://ir.ua.edu/bitstream/handle/123456789/9567/u0015_0000001_0004540.pdf?sequence=1).

### Line description

Both BY250 and UA44 strains were created using vectors inserted into the *C. elegans* Bristol N2 background (*Adams, 2022*). Specifically, in the BY250 line, neurons express Green Fluorescent Protein (GFP) after the integration of the pPdat-1::GFP construct into the genome (created by Dr. Randy Blakely, Florida Atlantic University; *Nass et al., 2005*). BY250 is a well-used model to explore the impact of neurodegeneration following chemical exposure from 6-hydroxydopamine and rotenone (*Nass et al., 2005*; *Ray et al., 2014*). The UA44 transgenic line was created at the University of Alabama (*Cao et al., 2005*) through coinjection of a plasmid with GFP and a second plasmid with human α-synuclein. When the two plasmids are coinjected into *C. elegans* they form stable extrachromosomal concatamers (*Mello et al., 1991*). We verified this using phenotype analysis where dopaminergic neurons, highlighted with GFP, displayed α-synuclein dose-dependent neurodegeneration as the animals aged (*Cao et al., 2005*; *Hamamichi et al., 2008*). Stably expressing worms were then chromosomally integrated to create UA44 (*Qiao et al., 2008*; *Kritzer et al., 2009*).

### DNA extraction and sequencing

Nematodes were grown to large population size on NGM plates seeded with *E. coli* OP50 (*Stiernagle, 2006*). Plates were washed with M9 media into 15mL conical tubes and rocked on a table rocker for an hour to purge any biological waste. The tubes were then centrifuged to pellet the worms and remove the supernatant, and then the pellet was washed with M9 approximately five times until the supernatant was clear to remove as much bacteria and waste from the worm pellet as possible. The pellet was then moved to a two mL tube and frozen at −20 °C until extraction. Genomic DNA was extracted with a modified phenol-chloroform extraction following *Sutton et al. (2021)*.

We used the ONT SQK-LSK109 ligation sequencing kit for library preparation with a modification, replacing the first AmpureXP bead clean with a treatment of the Short Read Eliminator Kit available from Circulomics Inc. Approximately 700 ng of gDNA were loaded onto a R9.4.1 RevD flow cell on the ONT GridION X5 platform and sequenced for 48 h. We performed standard base calling using Guppy v.4.0.11 and trimmed adaptor

sequences from DNA reads using Porechop (−−*discard_middle*; *Wick, 2018*). We used Nanoplot to calculate the read statistics after Porechop, and on the subset of reads used in the final Flye assembly (see assembly details; *De Coster et al., 2018*).

### Short-read sequence data

In order to polish the assembled DNA libraries, we downloaded Illumina paired-end DNA libraries from the NCBI Sequence Read Archive (SRA) in April 2020 (BioProject PRJDB2670; *NCBI Resource Coordinators, 2017*). In order to polish the sequences containing the transgene, we simulated 150 bp, 100x coverage paired-end DNA libraries for the insertion sequences with the software package ART (versionMountRainier; *Huang et al., 2012*). The simulated data was was added to the respective Illumina data downloaded from SRA and used to polish the completed genome assemblies with the software Pilon (described below; Pilon v1.23; *Walker et al., 2014*).

### Genome assembly, polishing, and scaffolding

We corrected raw ONT DNA libraries using the Canu –*correct* option, which corrects errors based on best overlaps among reads (Canu v1.9; *Koren et al., 2017*). The Canu-corrected reads were then assembled using Flye (v2.8.1; *Kolmogorov et al., 2019*). We used our paired-end short-read dataset containing the respective short-read libraries and simulated transgene insertion libraries to polish our draft assemblies four times with Pilon and eliminate small base pair errors and insertions/deletions (*Walker et al., 2014*) *Caenorhabditis* DNA libraries are frequently contaminated with foreign DNA (*Fierst & Murdock, 2017*; *Fierst et al., 2017*). To decontaminate our DNA libraries, we created a reference database for taxonomic assignment of contiguous sequences (contigs) with blastn using the NCBI 'nt' database (v2.2.31; *Altschul et al., 1990*). Only contigs aligning to *Caenorhabditis* were kept. Draft assemblies were then aligned, corrected, and scaffolded according to the genomic position within the *C. elegans* N2 reference genome (WBcel235) using RagTag (v2.0.1; *Alonge et al., 2022*). During the RagTag −−*correct* step we used the −*j* flag which allowed us to "hide" a list of query sequences with hits to the respective transgenes to prevent the module from removing the insertion due to lack of similarity to the surrounding *C. elegans* reference sequence. Six small sequence fragments under 1 kb were removed for final statistics and depositing with the NCBI.

### Transgene insertion identification

We aligned the vector insertion sequence for each assembled genome strain to the draft genome assembly before and after scaffolding using minimap2 (*Li, 2018*; *Li, 2021*). These locations were used to inform the correction step of RagTag to keep the insertion from being removed from the corrected reads. Additionally the insertion vector sequence was aligned to the final genome using blastn, to identify all stop and start locations for the insertion in each genome (*Altschul et al., 1990*). Annotated figures were made with gggenomes (*Hackl, 2022*).

### Quality assessment

The quality of the assemblies was assessed and compared to the *C. elegans* N2 reference genome (WBcel235) using QUAST (v5.1.0rc1; *Gurevich et al., 2013*). BUSCO (v5.3.2;

database: Nematoda_odb10) was used to assess the completeness of our genomes using a unique set of 3,131 genes expected to be conserved in a single copy within Nematoda (*Simão et al., 2015*; *Manni et al., 2021*). Both quality assessments were measured before and after scaffolding with RagTag (v2.0.1; *Alonge et al., 2022*).

### Annotation

Gene annotation was completed using the protein-coding sequences from *C. elegans* N2 (WBcel235) as a reference with the software Liftoff which is able to map annotated protein-coding sequences from one assembled genome to another when the two strains are closely related (*Shumate & Salzberg, 2021*). After annotation, the coding sequences (CDS) were extracted from the assembled genome sequence using the '*extract_sequences*' function from the AGAT package (*Dainat, 2020*), and a BED coordinates file was created from the GFF using the '*gff2bed*' function from BEDOPS (*Neph et al., 2012*). We used the WormBase Parasite BioMart (*Howe et al., 2017*) to extract identifying features associated with genes that were present in the *C. elegans* N2 reference protein-coding annotations but missing in our genome annotations. These features included orthologous and paralogous genes, chromosomal locations, gene ontology (GO) terms (*The Gene Ontology Consortium, 2000*) and Interproscan protein domain annotations (*Zdobnov & Apweiler, 2001*; *Jones et al., 2014*; *Finn et al., 2017*).

### Synteny analysis

The mRNA sequences and BED coordinates files generated during the annotation step were used to assess synteny with MCscan (python version) from the JCVI utility library (v1.1.17; *Tang et al., 2008*; *Tang et al., 2017*). We compared the UA44 and BY250 genome sequences to the standard *C. elegans* N2 (WBcel235) genome limiting our comparison to the six largest linkage groups representing the six *Caenorhabditis* chromosomes.

## RESULTS AND DISCUSSION

### Sequence data summary

After sequencing, the Oxford Nanopore DNA library for the UA44 strain contained 7.44 Gb of sequence reads with an N50 (the size median) of 4,732 bp with an average (mean) sequencing depth of 74.2x (Table 1). The BY250 library contained 12.45 Gb of sequence reads with an N50 of 6,116 bp with an average depth of 124.12x. The reads were then corrected with Canu –*correct* (*Koren et al., 2017*). The Canu-corrected UA44 DNA library was 3.88 Gb with an N50 of 8,082 bp with an average depth of 38.69x (Table 1). The Canu-corrected BY250 library was 4.10 Gb with an N50 of 15,109 bp and an average depth of 40.84x. Additional ONT library statistics can be found in Table S1.

### Assembly

The Canu-corrected reads were used to assemble both genomes with Flye (v2.8.1; *Kolmogorov et al., 2019*). The UA44 draft assembly contained 89 contigs with a length of 102.8 Mb (Table 2), and the BY250 draft assembly contained 118 contigs with a length

**Table 1 Read length statistics for UA44 and BY250 before and after correction calculated with the Nanoplot function from NanoPack (_De Coster et al., 2018_).** Canu-corrected (_Koren et al., 2017_) reads were used to assemble the genomes with Flye (_Kolmogorov et al., 2019_).

| Sequencing statistics | | | | |
|---|---|---|---|---|
| Line | Total bases (Gb) | Mean read length | N50 | Coverage |
| UA44 | 7.44 | 2,783.80 | 4,732.0 | 74.22 |
| Canu Corrected | 3.88 | 6,498.10 | 8,082.0 | 38.69 |
| BY250 | 12.45 | 3,070.00 | 6,116.0 | 124.12 |
| Canu Corrected | 4.10 | 11,885.10 | 15,109.00 | 40.84 |

**Table 2 Assembly statistics for UA44, BY250, and reference _C. elegans_ strains.** Assembly statistics for the _C. elegans_ reference (WBcel235), and before and after reference-based scaffolding statistics for UA44 and BY250.

| Assembly Statistics | | | | | | | | | |
|---|---|---|---|---|---|---|---|---|---|
| Assembly | Length (bp) | Contigs | Contigs 10,000 | Contigs 50,000 | N50 (Mb) | L50 | L90 | % elegans genome | GC% | Longest contig (Mb) |
| Reference | 100,286,401 | 7 | 0 | 6 | 17.5 | 3 | 6 | – | 35.44 | 20.1 |
| UA444 | 102,762,052 | 89 | 37 | 45 | 3.6 | 11 | 28 | 99.60 | 35.46 | 9.1 |
| Ragtag | 102,737,458 | 55 | 36 | 7 | 18.0 | 3 | 6 | 99.65 | 35.47 | 21.2 |
| BY250 | 102,451,217 | 118 | 14 | 94 | 1.8 | 18 | 57 | 99.45 | 35.46 | 5.9 |
| Ragtag | 102,462,217 | 46 | 19 | 7 | 17.9 | 3 | 6 | 99.44 | 35.46 | 21.1 |

of 102.5 Mb. After scaffolding with RagTag (v2.0.1; _Alonge et al., 2022_) and removing contigs under 1 kb, the UA44 assembly contained 55 contigs, and the N50, an assessment to measure the length of the scaffold that represents 50% of the genome, improved from 3.6 Mb to 18 Mb. The BY250 assembly contained 46 contigs after scaffolding, and the N50 improved from 1.8 Mb to 17.9 Mb. The scaffolded N50s of 18 Mb and 17.9 Mb for UA44 and BY250 respectively are comparable to the N50 of 18 Mb found in the reference assembly WBcel235. Additionally, the L90 for UA44 improved from 28 contigs to six, and from 57 contigs to six for BY250. With final L90s of six contigs and L50s of three contigs, our assemblies match those of the WBcel235 genome, indicating that correction and scaffolding improved the UA44 genome to near chromosome level. L90 represents the number of contigs that make up the first 90% of the length of the genome, and L50 represents the number of contigs that make up 50% of the length. Both genomes also show a consistent GC% of 35.47% matching the 35.44% of the reference. The final UA44 genome assembly represented 99.65% of the standard reference genome, and the final percentage match for BY250 was 99.44% (Table 2). These metrics indicate that our genome assemblies with reference-based scaffolding are very complete and near chromosome-level.

To assess the completeness of our genomes, we used BUSCO (v5.3.2) to search our genomes for a set of 3,131 orthologous genes present in nematodes (Nematode odb v10; _Simão et al., 2015_; _Manni et al., 2021_). The final assembly for UA44 showed a 98.9% BUSCO score matching the score of the reference (Table 3). UA44 had 21 missing genes, which is also the same as the _C. elegans_ reference. BY250 had a final BUSCO score of 98.6% with 24 missing genes, only three more than the reference. Over 97.9% of the genes were

Table 3  BUSCO (*Simão et al., 2015*; *Manni et al., 2021*) results for the *Caenorhabditis elegans* reference (WBcel235), and before and after reference-based scaffolding statistics for UA44 and BY250.

| BUSCO results | | | | | |
|---|---|---|---|---|---|
| Line | BUSCO | Single | Duplicated | Fragmented | Missing |
| Reference | 98.80% | 98.30% | 15 (0.5%) | 18 (0.6%) | 21 (0.6%) |
| UA444 | 98.70% | 98.00% | 23 (0.7%) | 18 (0.6%) | 21 (0.7%) |
| Ragtag | 98.80% | 98.10% | 21 (0.7%) | 18 (0.6%) | 21 (0.6%) |
| BY250 | 98.70% | 98.00% | 22 (0.7%) | 19 (0.6%) | 23 (0.7%) |
| Ragtag | 98.60% | 97.90% | 22 (0.7%) | 19 (0.6%) | 24 (0.8%) |

Table 4  Gene annotations for UA44 and BY250 compared to the *C. elegans* reference (WBcel235). Annotations were consistent with the reference with over 99.5% of genes present in the draft genomes. Additional $y > 99.8\%$ of the genes were found on the main six linkage groups or the mitochondria (I, II, III, IV, V, X, MtDNA).

| Gene annotation results | | | | | |
|---|---|---|---|---|---|
| Line | Genes | Missing | CDS | mRNA | % Genes in main linkage groups |
| Reference | 20,191 | – | 225,661 | 33,552 | – |
| UA44 | 20,100 (99.56%) | 91 (0.45%) | 225,239 (99.8%) | 33,451 (99.70%) | 99.8% |
| BY250 | 20,099 (99.54%) | 92 (0.46%) | 225,062 (99.74%) | 33,435 (99.65%) | 99.9% |

present in single copy in both genomes, very similar to the 98.3% of single copy genes in the reference. Our gene completeness results show that our genomes compare very well to the standard published *C. elegans* reference genome.

## Annotations

Annotations were mapped to the genome using Liftoff (*Shumate & Salzberg, 2021*). Results are shown in Table 4. Of the 20,191 genes present in the reference, 20,100 and 20,099 were lifted over to UA44 and BY250 respectively representing over 99.5% of the reference genes. Only 91 and 92 genes were not able to be mapped during liftover to UA44 and BY50. Of those 91 and 92 missing genes, 77 were shared between the UA44 and BY250 annotation. 46% of the genes missing in BY250 were located on Chromosome V in the *C. elegans* N2 reference annotations, 20% on Chromosome X, 10% on Chromosome II, 9% on Chromosome IV, 8% on Chromosome III and 8% on Chromosome I. UA44 missing genes had a similar chromosomal distribution (within 1–2%) to BY250.

Inspection of missing genes on Chromosome V of BY250 revealed two common patterns. First, 13 missing genes had no orthologous or paralogous genes, Gene Ontology (GO) (*The Gene Ontology Consortium, 2000*) or Interproscan protein domain annotations (*Zdobnov & Apweiler, 2001*; *Jones et al., 2014*; *Finn et al., 2017*). These genes were also very small for protein-coding genes, ranging in size from 20–36 nucleotides. Computational protein-coding genome annotation has a high false positive rate (*Bruna et al., 2021*) and it is possible these gene sequences were not accurate.

The second pattern we identified was missing genes that were verified in *C. elegans* and the result of complex patterns of molecular evolution. For example, the Seven TransMembrane Receptor gene *str-120* is part of a gene family that contains 21 orthologues

across *Caenorhabditis* and 256 paralogues within the *C. elegans* genome. Similarly, the EB1 C-terminal domain-containing protein *ebp-3* is part of a gene family that contains 158 orthologues but just two paralogous genes in *C. elegans*. *Caenorhabditis* genomes show rapid rates of gene family expansion and shrinkage through tandem duplications and deletions (*Adams et al., 2023*), leaving closely related complete and partial gene sequences that can be challenging for alignment and annotation (Figs. S1B, S2B). Overall, our coding sequence annotations (>99.7%) and mRNA annotations (>99.6%) were highly consistent with the reference genome. Additionall $y > 99.8\%$ of the annotated genes were contained on the six main linkage groups (I, II, III, IV, IV, X) or the mitochondria.

## Transgene insertion

Prior to scaffolding, the vector insertion sequence for UA44 aligned to Contig59 which then scaffolded into Chromosome IV. For BY250 the vector insertion sequence aligned to Contig86 which then scaffolded into Chromosome I. The BY250 and UA44 transgenes mapped to expected genomic regions based on outcomes from genetic crosses with these transgenes (*Gaeta et al., 2023*). These locations were used to exclude Contig59 and Contig86 from the RagTag (*Alonge et al., 2022*) correction step of scaffolding for their respective genomes.

We also used blastn to align the vector sequence to the genomes (*Altschul et al., 1990*). For UA44, the vector insertion was found in four complete copies on Chromosome IV. The locations of the 4,924 base pair vector were: IV:8767911-8763011 (−4,900 bp), IV:8772965-8768045 (−4,920 bp), IV:8777841-8772966 (−4,875 bp), and IV:8784656-8789078 (4,422 bp) (Table 5). Three of the locations were negatively oriented, while one was positively oriented (Fig. 1, created with gggenomes; *Hackl, 2022*). Additional partial hits of the insertion vector sequence to the genome were found, however, none of these partial hits contained the $\alpha$-synuclein gene (Table S2). The four-copy insertion of $\alpha$-synuclein into Chromosome IV of UA44 may increase the overall expression of the genes present in the insertion. Parkinson's severity in humans shows copy-number dependence including within-family genomic triplication of SNCA (the $\alpha$-synuclein locus) (*Singleton et al., 2003*; reviewed in *Cognata et al., 2017*). Importantly, annotations of the surrounding areas indicate that the four-copy insertion did not affect genes in that region (Fig. 1, created with gggenomes *Hackl, 2022*). Some of the partial vector insertions either overlapped or were adjacent to genic sequences (Table S2). These intersections may be potentially capable of altering the regulation of proteins or gene expression.

For the BY250 transgene, only two BLAST (*Altschul et al., 1990*) hits were found. The primary insertion location was on chromosome I:11058905–11058020 and was 885 bp of the 1,113 bp insertion vector sequence (Table 5). A second partial insertion hit was also found on chromosome I:11059035–11058951, but was only 84 base pairs long. The BY250 insertion found in single copy on chromosome I did not intersect any surrounding genes (Fig. 2, created with gggenomes *Hackl, 2022*).

## Synteny

Coding sequences and gene BED coordinates were used to align each genome to the *C. elegans* N2 (WBcel235) reference genome using MCscan (Python version) to assess synteny

Adams et al. (2024), *PeerJ*, DOI 10.7717/peerj.18100

**Table 5** **Insertion location information for UA44 and BY250.** Insertion location information for UA44 and BY250. UA44 has four near complete insertions located on Chromosome IV, while BY250 contains only one complete insertion on Chromosome I. Additionally, none of the insertions intersect the surrounding genes.

| | | | Insertion locations | | | | | | | | | | | |
|---|---|---|---|---|---|---|---|---|---|---|---|---|---|---|
| Line | Sequence | | Location | | | Insertion | | | Blast score information | | | | | |
| | ID | Length | Start | Stop | Length | Start | Stop | Length inserted | Match % | Matches | Length | Gaps | Bit score | Evalue |
| | | | 8772965 | 8768045 | (-)4920 | 1 | 4924 | 4923 | 99% | 4897 | 4945 | 45 | 8824 | 0 |
| | | | 8767911 | 8763011 | (-)4900 | 18 | 4924 | 4906 | 99% | 4876 | 4928 | 48 | 8769 | 0 |
| UA44 | IV | 17.96 Mb | 8777841 | 8772966 | (-)4875 | 1 | 4924 | 4923 | 99% | 4865 | 4925 | 50 | 8717 | 0 |
| | | | 8784656 | 8789078 | 4422 | 1 | 4406 | 4405 | 99% | 4399 | 4428 | 27 | 7991 | 0 |
| BY250 | I | 15.37 Mb | 11058905 | 11058020 | (-)885 | 228 | 1113 | 885 | 99% | 875 | 886 | 0 | 1576 | 0 |
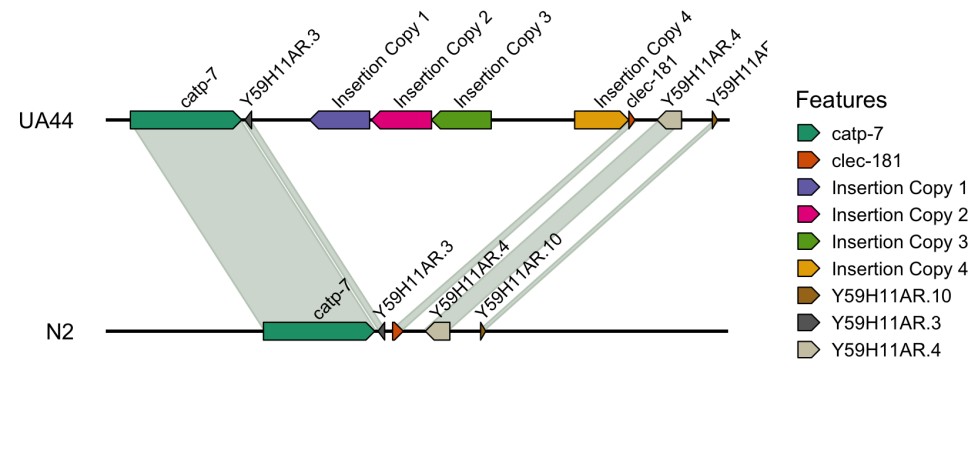

**Figure 1 Insertion locations for UA44.** Four copies of the insertion vector were found on Chromosome IV: copy 1 = IV:8767911-8763011, copy 2 = IV:8772965-8768045, copy 3 = IV:8777841-8772966, and copy 4 = IV:8784656-8789078. Genes flanking the insertion region and the alignment to the N2 genome are shown.

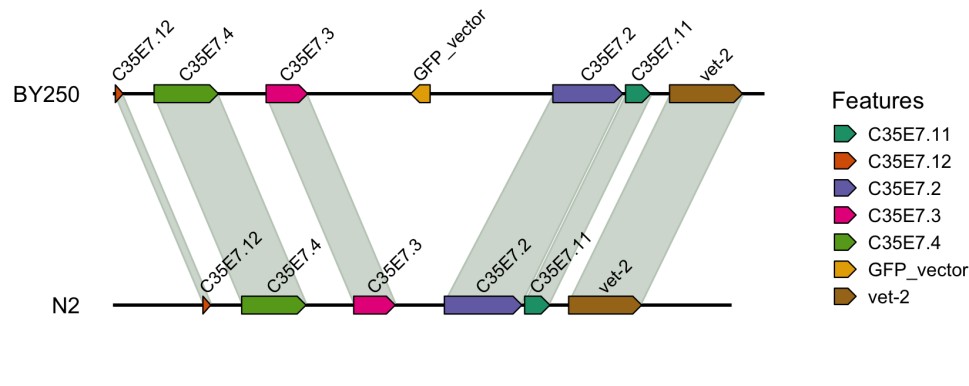

**Figure 2 Insertion locations for BY250.** One copy of the insertion vector was found on chromosome I:11059035-11058951. Genes flanking the insertion region and the alignment to the *C. elegans* N2 genome are shown.

(*Tang et al., 2008*; *Tang et al., 2017*). Macrosynteny visualization shows our genomes covering the seven chromosomes of the *C. elegans* N2 reference (Fig. 3). Macrosynteny plots for each assembly aligned to the reference along with syntenic maps and alignment depth comparisons are available in the Figs. S1 and S2.

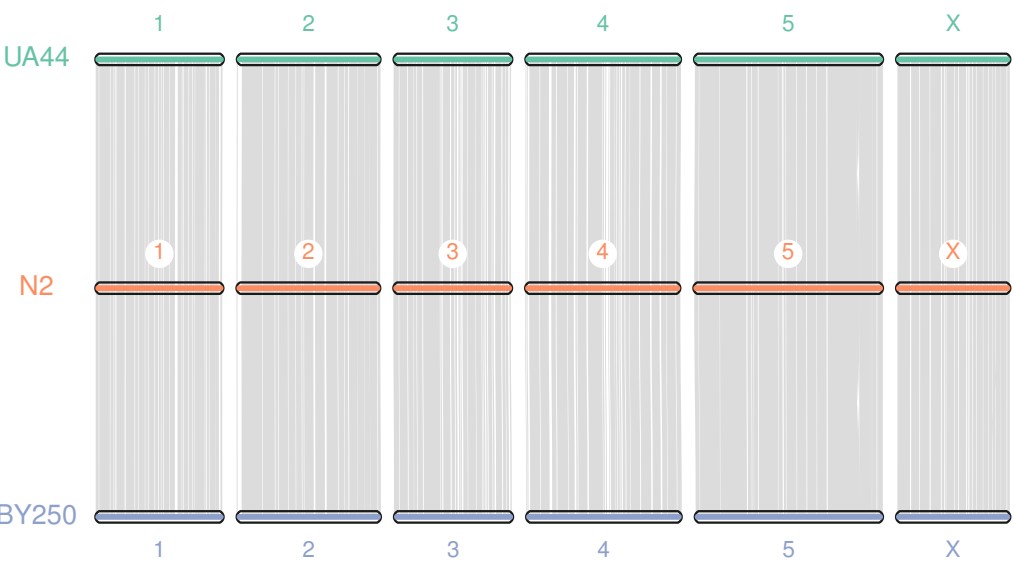

**Figure 3** **Macrosynteny alignment for BY250, N2, and UA44.** Macrosynteny alignment for the BY250 and UA44 genomes to the N2 *C. elegans* reference genome.

## SUMMARY

The physical processes that create transgenic insertions result in random incorporation of the transgene into the nuclear DNA of the target organism. Previously, these locations were difficult to identify and characterize. We have successfully sequenced two transgenic lines of *C. elegans* using ONT and identified the exact size, location and frequency of insertions. Our results add to a body of literature (*Suzuki et al., 2020*; *Nicholls et al., 2019*; *Giraldo et al., 2021*; *Jain et al., 2016*) demonstrating that long-read sequencing allows for rapid, cost-effective, high-quality genome assembly and facilitates identification of transgene insertions.

Genome sequencing and reference genome assembly have been pursued analogous to museum collections where a single holotype specimen was used for species description and designation (*Gong et al., 2023*). Population sequencing has made it clear that genomic diversity is ubiquitous and there is a pressing need to describe the pangenome, the full complement of genomic diversity spanning a species (*Miga & Wang, 2021*). For example, across *C. elegans* wild-collected strains genome size varies by 2–8% despite the worms remarkably similar phenotypes (*Thompson et al., 2015*; *Kim et al., 2019*). Lab adaptation and drift additionally produce mutations, segregating differences and genomic differentation in short timescales (*Bush et al., 2024*).

*C. elegans* strains include mutations both natural and engineered. Mutant lines have been created for over 50 years as genetic tools through aggressive methods like X-rays, UV and gamma radiation, often used to cause large-scale structural variations and balancer strains unable to recombine (*Nigon & Félix, 2005-2008*). These have been used in laboratory experiments for generations but only recently studied with whole genome sequencing (*Maroilley et al., 2023*). Even this study used only short read Illumina sequences and
acknowledged that without long ONT or Pacific Biosystems DNA libraries many large-scale structural variants, including the causal blocks to recombination, remain undiscovered. The experimental and bioinformatic methods we have presented here could be used to readily study these and other fundamental genetic tools.

Our study prioritized feasibility, including both sequencing costs and human and computational time. Despite the promise of ONT and Pacific Biosystems long read platforms, multiple requirements leave high-depth, high-quality, ultra long DNA sequence reads out of reach for much of the global scientific community. ONT is highly sensitive to variations in organismal input, DNA extraction and library preparation (*Jain et al., 2016*). The cost associated with both ONT and Pacific Biosystems sequencing is prohibitive given science funding in most countries.

Our assembled genome sequences were highly contiguous, and scaffolding using the reference genome allowed us to achieve genome sequences approaching chromosome-level assemblies. Despite the precision and length of Pacific Biosystems and Oxford Nanopore Technologies DNA libraries, Hi C and other technologies are often required to assemble full chromosomes in *C. elegans* (*Tyson et al., 2018*; *Lee et al., 2021*). We found that reference-based scaffolding with RagTag (*Alonge et al., 2022*) required a few discrete alterations to ensure the transgene, a foreign DNA sequence, was not edited out of the assembled sequence. After we performed these, we were able to achieve highly contiguous, chromosome-scale sequences. BUSCO (*Simão et al., 2015*; *Manni et al., 2021*) scores over 98% along with annotation of >99% of *C. elegans* genes, further supports high genomic integrity in our assembled sequences. The location of the transgene insertions was easily identifiable with alignment with minimap2 (*Li, 2018*; *Li, 2021*) and local BLAST (*Altschul et al., 1990*) alignment thanks to long ONT sequence reads spanning the insertion site.

We identified four complete copies of the insertion vector on Chromosome IV in the UA44 strain (Fig. 1) and multiple partial hits to the vector sequence. In comparison, only one copy of the transgene was found on Chromosome I of BY250 (Fig. 2). The multiple copies of the $\alpha$-synuclein insertion present in UA44 may result in increased gene expression of the genes present in the vector, and may have implications for gene expression of the surrounding genomic regions. The SNCA locus that produces the $\alpha$-synuclein protein in humans is present in multiple copies and Parkinson's severity increases with copy number (reviewed in *Cognata et al., 2017*). The four complete copies that inserted into the *C. elegans* genome mirror natural variation in humans and mimic the most severe Parkinson's-causing triplication found in humans (*Singleton et al., 2003*).

Our results highlight the complications that may arise between laboratory mutational process and realized genomic mutation in transgenic creation. Even more precise transgenic techniques have the potential to, as in this study, insert multiple times. The location and genomic context of the transgenic insertion can also influence the genome and organism. Transgenes can insert into other genes, promoters, non coding RNAs or regulatory sequences (*Nigon & Félix, 2005-2008*). Recent technical developments like Hi-C permit study of topological association domains and physical relationships but the influence that disrupting these physical relationships has on gene regulation and expression is

not understood (*Akdemir et al., 2020*; *Allou & Mundlos, 2023*). Precise characterization is necessary to discover these relationships.

*C. elegans*, unlike many organisms, is a self-fertile hermaphrodite and individuals can be frozen at ultra low temperature and revived (*Stiernagle, 2006*). These procedures ensure transgenes are subject to minimal mutational processes once inserted. However, in other model organisms transgenes are subject to continual mutational pressures including single nucleotide changes, duplications, insertions and even total deletion (*Suzuki et al., 2020*). Utilizing genome sequencing and assembly to characterize and locate transgenes can provide insight into the mutation-altered states the insertions acquire over time.

Our study demonstrates that even with the massive resource base available for *C. elegans*, care must be taken when assembling transgenic lines to ensure correct assembly and scaffolding. The creation of transgenic lines may affect the integrity of genes surrounding any incorporation of the transgene; however, we find no evidence of interrupted genes near the insertion locations. We hope these assembled genome sequences will be a great resource for the worm community, and that our study outlines a viable method for identifying the genomic basis of engineered mutations.

## ACKNOWLEDGEMENTS

We gratefully acknowledge helpful discussion, feedback, and comments from Karolina Willicott and other members of the Fierst and Caldwell labs. We thank Randy Blakely for creating the BY250 mutant strain and Laura Berkowitz for creating the UA44 mutant strain.

### Funding

Paula E. Adams and John M. Sutton were supported by National Alumni Association Fellowship through the University of Alabama Alumni Association. Jennifer L. Thies was supported by NIA award R36AG073798, KAC was supported by NINDS R15NS106460 and GAC was supported by NINDS award R15NS104857. Janna L. Fierst, Paula E. Adams, John M. Sutton and Joshua D. Millwood were supported by National Science Foundation grants EF-1921562 and DEB1941854 and Janna L. Fierst was supported by NIGMS award R35GM147245. The funders had no role in study design, data collection and analysis, decision to publish, or preparation of the manuscript.

### Grant Disclosures

The following grant information was disclosed by the authors:
National Alumni Association Fellowship through the University of Alabama Alumni Association.
NIA award: R36AG073798.
KAC: NINDS R15NS106460.
GAC: NINDS award R15NS104857.

National Science Foundation grants: EF-1921562, DEB1941854.
NIGMS award: R35GM147245.

## Competing Interests

The authors declare there are no competing interests. John M. Sutton is employed by Absci.

## Author Contributions

- Paula E. Adams conceived and designed the experiments, performed the experiments, analyzed the data, prepared figures and/or tables, authored or reviewed drafts of the article, and approved the final draft.
- Jennifer L. Thies performed the experiments, authored or reviewed drafts of the article, and approved the final draft.
- John M. Sutton performed the experiments, authored or reviewed drafts of the article, and approved the final draft.
- Joshua D. Millwood performed the experiments, authored or reviewed drafts of the article, and approved the final draft.
- Guy A. Caldwell conceived and designed the experiments, authored or reviewed drafts of the article, and approved the final draft.
- Kim A. Caldwell conceived and designed the experiments, authored or reviewed drafts of the article, and approved the final draft.
- Janna L. Fierst conceived and designed the experiments, prepared figures and/or tables, authored or reviewed drafts of the article, and approved the final draft.

## Data Availability

The sequencing data associated with this article are available at National Center for Biotechnology Information: PRJNA627736 (UA44) and PRJNA627737 (BY250).

The Illumina 300 bp paired-end sequencing library used for polishing is available at BioProject: PRJDB2670, DRX007632.

The bioinformatic scripts and workflow are available at GitHub and Zenodo:

- https://github.com/peadams/IdentifyingTransgene_Celegans

- Adams, P. (2024). Adams et al. Identifying transgene insertions in Caenorhabditis elegans genomes with Oxford Nanopore sequencing Resources [Data set]. Zenodo. https://doi.org/10.5281/zenodo.13359289.

## Supplemental Information

Supplemental information for this article can be found online at http://dx.doi.org/10.7717/peerj.18100#supplemental-information.

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
