# Peer review of "Identifying transgene insertions in Caenorhabditis elegans genomes with Oxford Nanopore sequencing"

_PeerJ, doi:10.7717/peerj.18100_

## Round 0.1 · original submission · Minor Revisions

Please add the information requested by Reviewer 1, and the article should be ready to go.

Reviewer 1 ·

Basic reporting

This is a report of two genomes of C. elegans strains. Both are transgenic strains used for as models for neurodegenerative disease with specific insertions that have been generally but not precisely localized. In this report, the authors de novo assemble the strains to a chromosome-level using new relatively high-depth Oxford Nanopore sequencing and existing short read sequencing for polishing. On a whole, suitable methodologies for genome assembly are used and reporting is consistent with other genome papers. I’m requesting some additional information about potential gene annotations (if genes are missing that seems relevant to a study reporting on these strains) and have some suggestions about the basic results reporting.
Line 75-78: “For example, of the 24,000 mutant lines maintained by the CGC the genomes of only 2 - the strains reported in this study - have been sequenced, characterized at a whole genome level, and deposited with the National Center for Biotechnology Information (NCBI) for use by the scientific community.”
This part of the introduction could be read to imply that the 2 strains you report in this study have already been sequenced/characterized/deposited/published prior to this study. If you are saying that this paper is the first report of these two assemblies perhaps rephrase to clarify.
Line 194-197” Of the 20,191 genes present in the reference, 20,100 and 20,099 were lifted over to UA44 and BY250 respectively representing over 99.5% of the reference genes. Only 91 and 92 genes were not able to be mapped during liftover to UA44 and BY50. Of those 91 and 92 missing genes, 77 were shared between the UA44 and BY250 annotation. “
Do these 77-92 genes appear to be clean deletions, or rather frame-shifting or other disabling mutations relative to the reference? It is possible the reference annotation is incorrect and these are not actually good annotations and that would be interesting to know. Alternatively, they may not have assembled at all due to being in difficult assemble regions. This could show up as the genes being found in gaps between contigs in the new assembly. Since there are very few genes, it should not be too difficult to check by hand at least a random subset and see what may be going on. It’s also relevant to the overall purpose of the experiment to characterize these strains in detail to be sure phenotypic differences are due only to the transgenic nature of the strain and not other unknown differences.
Results Paragraph starting on Line 200:
1. Are “insertion sequences for each genome” referenced in the first paragraph the same sequences as the “vector insertions” referenced later, and the difference here is whether minimap2 or BLAST was used to align them? Or are these different sequences? The methods appear to indicate that minimap2 step was done only to exclude contigs to avoid filtering them and BLAST was the primary tool used to identify the exact coordinates and copy number of the same sequences. I wonder if confusion could be avoided if the minimap results were included in the methods instead and this paragraph just focus on the BLAST identification of the insertion locations and copy number. Otherwise perhaps some rephrasing would help to clarify.
2. The existence of additional partial vector insertions were mentioned, but it’s not discussed whether these “off-target” insertions could disrupt genes, which seems relevant. The supplement doesn’t seem to contain this information either, though it does list the coordinates. At least the number of partial insertions on each chromosome for UA44 should also be mentioned in the text.
Figure 3 – I don’t find this figure too helpful as they are so similar (appear by eye to be 100% syntenic?). Can you do something to highlight the few differences that do exist? You should be able to make the lines darker or colored for specified locations for example. I’m not sure about how to do it in gggenomes but it’s possible in a regular ggplot2 plot (geom_segment). Maybe three dotplots could be used?
Figure S1-S2 appear blurry on my pdf reviewer copy. Its’ difficult to see the dots on the dotplot for example (there are very few since the genome’s mostly syntenic to the reference, so this is more important than it might usually be).
The dotplots for S1 and S2 appear to be identical (not sure due to blurryness). Could this be because the N2 reference is actually biologically different from the two new genomes and/or misassembled? Otherwise it would imply the same “errors” have occurred in both your two assemblies which seems unlikely.

Experimental design

No comment

Validity of the findings

Line 204-205 “The BY250 and UA44 transgenes mapped to expected genomic regions based on outcomes from genetic crosses with these transgenes (Caldwell lab; unpublished observations)”

It would be in line with the PeerJ policy of including "all underlying data" to include a summary of these observations as a supplement, given the unpublished observation is from one of the Authors. Detailed raw data are likely not necessary, but something about the strains used and what was consistent in the outcome would be helpful.

Reviewer 2 ·

Basic reporting

Adams et al., in this manuscript do a splendid job of assembling high quality genomes of two C. elegans transgenic lines that are commonly utilized as models in biomedical studies. The manuscript is well written, with sufficient detail and rigor.

Experimental design

Details of the long read sequencing, assembly, annotation, synteny, and quality assessment are reported with care, and represent some state-of-the-science methods. The research questions are well presented, with great motivation for the work, considering that these transgenic lines are used extensively across numerous fields. Considering that these lines still don't have detailed genome assemblies, this work has great potential to be useful to the worm community at large. I specifically appreciate the attention to detail as the authors present all bioinformatics scripts used + data on their GitHub page.

Validity of the findings

The quality of both assemblies is excellent, and will definitely provide an important resource to the C. elegans community.

---

## Round 0.2 · accepted · Accept

I confirm that this paper is ready for publication.